

# Assessing the effect of heavy metals on maize (*Zea mays* L.) growth and soil characteristics: plants-implications for phytoremediation

Muhammad Imran Atta[1], Syeda Sadaf Zehra[1], Habib Ali[2], Basharat Ali[2], Syed Naveed Abbas[1], Sara Aimen[3], Sadia Sarwar[1], Ijaz Ahmad[4], Mumtaz Hussain[5], Ibrahim Al-Ashkar[6], Dinakaran Elango[7] and Ayman El Sabagh[8,9]

[1] Department of Botany, The Islamia University of Bahawalpur, Punjab, Bahawalpur, Pakistan
[2] Agricultural Engineering, Khwaja Fareed University of Engineering and Information Technology, Rahim Yar Khan, Pakistan
[3] Department of Botany, the Women University, Multan, Pakistan
[4] Faculty of Agricultural Sciences, Bahauddin Zakariya University Multan, Multan, Pakistan
[5] Department of Veterinary Sciences, The Islamia University of Bahawalpur, Punjab, Bahawalpur, Pakistan
[6] Department of Plant Production, College of Food and Agriculture Sciences, King Saud University, Riyadh, Saudi Arabia
[7] Department of Agronomy, Iowa State University, Ames, USA
[8] Department of Agronomy, Faculty of Agriculture, Kafrelsheikh University, Kafr al-Sheik First, Egypt
[9] Department of Field Crops, Faculty of Agriculture, Siirt University, Siirt, Turkey

Corresponding authors
Muhammad Imran Atta,
imranbotany80@yahoo.com
Habib Ali, habib.ali@kfueit.edu.pk

## ABSTRACT

**Background**. Heavy metal pollution has become a global environmental issue. Heavy metals are contaminating the agro-soils, growing crops, and vegetables through different agricultural practices. In this study, besides the phytoremediation potential of maize, the role of chromium (Cr) and lead (Pb) on crop and soil health has been investigated.

**Methods**. Two maize varieties, Pak-Afgoi and Neelem, were grown under varying concentrations of Cr (50–300 ppm) and Pb (30–300 ppm) and different growth parameters *i.e.*, seed germination, leaf size/number, stem girth, plant height, biomass, chlorophyll content, relative growth rate (RGR), and net assimilation rate (NAR) were studied under Cr and Pb stress. Likewise, the effect of metals was also assessed on different soil characteristics including soil texture, pH, EC, soil organic matter, urease activity and nutrients.

**Results**. Studied plant attributes were adversely affected by heavy metals toxicity. Affected values of RGR and NAR showed a linear correlation with affected growth and dry matter yield of maize. Heavy metals impacted different soil parameters including soil microbial performance and revealed a declining trend as compared to control soil. Maize varieties showed a significant phytoremediation potential *i.e.*, uptake of Cr and Pb was 33% and 22% in Pak-Afgoi, while Neelem showed 38% and 24% at 300 ppm, respectively. Data regarding metal translocation factor (TF), bioaccumulation factor (ACF), and biomagnification ratio (BMR) significantly revealed the potential of maize varieties in the removal of Cr and Pb metals from affected soils. However, Cr-accumulation was higher in shoots, and Pb accumulated in plant roots showed a differential behavior of metal translocation and affinity with the varieties. These maize

varieties may be recommended for general cultivation in the Cr and Pb-contaminated areas.

## INTRODUCTION

Abiotic stresses including metal toxicity, salinity, temperature extremes, soil microplastic, and drought are enormous threats that are affecting agriculture and the natural environment (*Wang, Vinocur & Altman, 2003*; *Hasanuzzaman et al., 2020*; *Li et al., 2023*). Heavy metals are one of the abiotic factors and are demarcated as metals with a density above 5 g/cm$^3$ *e.g.*, chromium (Cr), lead (Pb), nickel (Ni), cobalt (Co), arsenic (As) and silver (Ag), *etc.* These heavy metals diverge in physical and chemical properties; and are taken as substantial environmental pollutants owing to their toxic interaction with soil properties, plants, animals, and humans (*Hasan et al., 2009*; *Das & Jayalekshmy, 2015*; *Zhang et al., 2019*; *Karkush & Ali, 2020*). An incessant boost of heavy metals in the agricultural soil system is over diverse agricultural practices including the use of industrial and sewage waste-waters as a crop irrigation source. Irrigation of vegetables and fodder crops with such a kind of wastewater through their discharge into freshwater bodies is a common practice (*Khan, Khan & Aslam, 2003*); and hence is the foremost source of heavy metal pollution for intact growing crops in the peri-urban areas (*Mussarat, Bhatti & Khan, 2007*).

Chromium (Cr) is a toxic pollutant and is ranked as the 17th most toxic element among the hazardous substances (*Wakeel, Xu & Gan, 2020*). Its density is 7.15 g/cm$^3$ and ranged from 10 to 50 mg/kg of soil, naturally (*Kouser & Khan, 2021*). Chromium is used in electroplating, textile dying, paint, metallurgy, pigment, and tanning industry. Similarly, sewage and fertilizers are also included as the main source of Cr (*Amin et al., 2013*). Due to different oxidation states, Cr acts as a toxic element for organisms. However, Cr-toxicity to plants depends on its uptake mechanisms, concentration, and focal plant species (*Gardea-Torresdey et al., 2005*; *Peternella, da Silva & da Costa, 2021*).

Lead (Pb) is ranked as the second most toxic metal on earth's crust and is toxic to humans and other living things including plants. Its density is 11.34 g/cm$^3$ (*World Health Organization, 2010*). Several industrial processes include Pb-use in their products like oil and paint, mines, agrochemicals, *etc.* Moreover, Pb as salts or oxides is also being added to the environment through atmospheric dust, and automobile exhaust (*IARC, 2012*; *Kumar et al., 2019*). In nature, Pb remains below 50 mg kg$^{-1}$, but in some plants, Pb usually inhibits the growth mechanism when it is at a concentration of 30 mg/kg or more (*Usman et al., 2020*), while some of the plant species can tolerate Pb stress up to 1,000 mg kg$^{-1}$ (*Reeves et al., 2018*). Both Cr and Pb have a strong effect on different growth attributes of

exposed plants (wheat, maize, barley, sunflower, mustard, and soybean); and inhibit seed germination, plant height, root-shoot length, fresh-dry weight of seedlings, tolerance index, leaf number and photosynthesis (*Orhue & Ekhomun, 2010*; *Naseem et al., 2015*; *Akhtar & Iram, 2017*; *Kanwal et al., 2020*).

Changes in soil properties depend upon the mobility and chemical activity of heavy metals in the soil predominantly when these metals exceed the accepted limits (*Karkush, Zaboon & Hussien, 2014*; *Uddin, 2016*; *Karkush & Ali, 2020*). Metal ions have acidification effects on the intact soil and lower the pH of the soil (*Motuzova, Makarichev & Petrov, 2011*). Soil pH is an important parameter that significantly affects the accessibility of soil nutrients available to the growing crops, affects their yield, and hence, acts as the key factor in sustainable agriculture (*Ludwig et al., 2001*; *Najafi & Jalali, 2016*). Soil respiration ($CO_2$ evolution) is an indicator of the use of energy by soil microbes concerning their efficiency in degrading the soil organic material (*Wardle & Ghani, 1995*).

In agricultural management, microbes are taken as soil indicators for the affecting external abiotic stresses including heavy metals (*Hassan et al., 2013c*), and are quite sensitive to such stresses. Microbes release important extracellular enzymes in the soil system, which are the key regulators of soil biochemical processes (*Wang & Yanli, 2013*). Soil urease is one of the most concerning extracellular enzymes released by microbes to hydrolyze soil urea into $CO_2$ and ammonia (*Gulser & Erdogan, 2008*). Similarly, soil enzymes act as biological catalysts and facilitate different soil reactions and metabolic processes of the biogeochemical cycles of soil nutrients to maintain soil fertility for growing crops (*Moreno, Garcia & Hernandez, 2003*). As the heavy metals put adverse effects on soil properties, some efficient and cost-effective techniques are needed to restore the metal-affected agro-soils. Phytoremediation is a biological remediation technique that has received a lot of attention during the last few years. Nevertheless, plant efficiency for phytoremediation depends upon the type, availability, and concentration of heavy metals. Phytoremediation easily removes metal contaminants from the affected soil than other remediation options (*Marques, Rangel & Castro, 2009*). It increases soil fertility through the release of different organic matter (from plant body) and hence, maintains the physical and biological properties of the soil (*Aken, Correa & Schnoor, 2009*; *Wuana & Okieimen, 2011*; *Jacob et al., 2018*). Plants used in the phytoremediation of heavy metals may be hyper-accumulator or phyto-stabilizer. Family Brassicaceae (*Alyssum bertolonii*; *Thlaspi caerulescens)* and Asteraceae (*Calendula o ffi cinalis*; *Tagetes erecta*) have a greater hyper-accumulating ability (*Glick, 2012*). Similarly, the phytoextraction potential of soybean (*Glycine max* L.) and rice (*Oryza sativa* L.) for phytoextraction of cadmium-polluted lands has also been reported by *Murakami, Ae & Ishikawa (2007)*. Moreover, *Lolium perenne, Panicum aquaticum, Typha species, Vetiveria zizanioides, and Paspalum fasciculatum* have also been documented as good phytoremediation tools for Cd, Cu, As, Zn, Cr, and Pb (*Glick, 2012*; *Alvarenga et al., 2009*; *Andra et al., 2009*; *Dipu, Kumar & Thanga, 2012*; *Pires-Lira et al., 2020*).

Data were collected as previously described by *Atta et al. (2023)* that due to excessive irrigation with wastewater, Cr and Pb have become the most frequent and health risk metals to the consumers of the study area (Dera Ghazi Khan, Punjab-Pakistan). Similar findings about heavy metal pollution in Dera Ghazi Khan have been reported by *Rafique et*

*al. (2016)*. To our knowledge, the toxicity of Cr and Pb on soil physicochemical properties and soil enzyme activity under maize cultivation has not been documented adequately in the study area. Therefore, a pot experiment was set to understand the effect of heavy metals not only on *Zea mays* seedlings but also on different soil characteristics. Moreover, this study also uncovered the phytoremediation potential of maize to combat the heavy metal issue in the future under the particular environmental conditions of the area. The maize crop is exceptionally grown as a fodder crop in this study area; therefore, maize has been selected for the current phytoremediation study.

## MATERIALS & METHODS

### Experimental design, germination and growth attributes

To evaluate the metal toxicity and phytoremediation potential of maize varieties, separate Petri plate and pot experiments were conducted in a block design during July–September 2018–20 (Temp. 35 °C–32 °C, Humidity 56%–59%). Open-pollinated varieties (OPV) of maize (Pak-Afgoi & Neelem Desi) were used in the trial. Potassium dichromate and lead nitrate ($K_2Cr_2O_7$) & $Pb(NO_3)_2$ were used and a stock solution was prepared *viz.* Cr (50, 100, 150, 250, 300 ppm) and Pb (30, 60, 100, 150, 300 ppm).

Each treatment comprised of eight replicates followed by three maize plants per treatment.

A seed germination test was performed in the laboratory at room temperature (30 °C). To prevent fungal infection during the experiment, the selected seed material was thoroughly washed with 2% sodium hypochloride for 5 min and then rinsed with distilled water. Seeds were imbibed in distilled water for 30 min and then were air-dried. For either variety, each Petri plate (10 cm in diameter) was employed with two filter papers and 10 seeds following four replicates per treatment. Each Petri plate was moistened with 10 mL of the metal solution while the control treatment continued with distilled water. Overall, the plates were observed daily for moisture/treatment requirements. Germinated seeds with one mm radicle were counted daily till the final germination day (day 10). Percent seed germination was determined following the study of *Akinci & Akinci (2010)* using the formula:

(Germination (%) = Total seeds germinated / total seeds arranged × 100).

Pre-washed, cleaned, dried, and labeled plastic pots of varying identifiable colors (dimension (cm): 30.5 diameter × 46 deep) were smoothly filled with 12 kg of the agro-soil (clay 63.1%, sand 29.7%, saturation 54%, EC (mS/cm) 2.8, pH 7.6, SOM content 2.7%, available-P 8.2 ppm, available-K 182 ppm, and 2.8% N). For either variety, healthy seeds of uniform size were sown 1–2 inches deep in the topsoil of the pots. After ten days of establishment, the seedlings were thinned by removing weak seedlings, and metal treatment was simulated for up to four weeks. Different growth attributes *i.e.*, plant length, leaf area (*Aliu, Fetahu & Rozmam, 2010*), SPAD value, plant fresh and dry weight, relative growth rate (RGR), and net assimilation rate (NAR) were assessed for the varieties at Harvest-1 and Harvest-2 *i.e.*, 25th and 40th day of growth (Table 1). RGR and NAR were assessed by using the method of *Causton & Venus (1981)* by the given formula:

$$RGR = Log_e w_2 - Log_e w_1 / t_2 - t_1$$

**Table 1** Comparison of growth at harvest-2 in Pak-Afgoi and Neelem under Cr & Pb treatment.

| SOV | Cr (ppm) | | | | | | Pb (ppm) | | | | | F-value | LSD (5%) |
|---|---|---|---|---|---|---|---|---|---|---|---|---|---|
| Control | 0 | 50 | 100 | 150 | 250 | 300 | 30 | 60 | 100 | 150 | 300 | | |
| **Variety: Pak-Afgoi** | | | | | | | | | | | | | |
| Germination (%) | 97.5 | 97.5 | 86.5 | 75 | 62.6 | 31 | 90 | 82.5 | 62.5 | 52.5 | 35.8 | 41.8 | 7.19 |
| Plant height (cm) | 68 | 68.4 | 66.4 | 62.1 | 58.9 | 53.3 | 69 | 66.2 | 61.7 | 57.3 | 49.7 | 4.86 | 2.57 |
| Leaf area (cm$^3$) | 74 | 74 | 72 | 69.5 | 67.5 | 65.5 | 72 | 70 | 67.8 | 67.5 | 63.2 | 3.56 | 0.79 |
| Leaf area (cm$^3$): 25d | 31.4 | 32 | 30.4 | 27.8 | 22.2 | 22 | 31 | 30 | 27.8 | 26.2 | 24.8 | 2.65 | 1.45 |
| Green leaf count | 10 | 10 | 10 | 9 | 9 | 9 | 10 | 9.5 | 9.5 | 9 | 9 | 0.41[NS] | 0.17 |
| Fresh weight (g) | 46.2 | 45.8 | 45.5 | 43.7 | 41.4 | 40.4 | 45.9 | 44.8 | 42.6 | 39.9 | 38.4 | 2.7 | 1.42 |
| Dry weight (g) | 17 | 17 | 14.5 | 13 | 12.2 | 10.5 | 17 | 14 | 12.8 | 12.1 | 10 | 3.46 | 0.47 |
| Dry weight (g): 25d | 7.4 | 7.4 | 7.4 | 6.5 | 6.2 | 5.5 | 7.4 | 7.4 | 6.2 | 6.2 | 5.3 | 2.18 | 0.23 |
| RGR (g day$^{-1}$) | 0.64 | 0.6 | 0.5 | 0.4 | 0.38 | 0.32 | 0.64 | 0.47 | 0.44 | 0.37 | 0.31 | 3.05 | 0.03 |
| NAR | 0.29 | 0.29 | 0.28 | 0.24 | 0.24 | 0.23 | 0.31 | 0.27 | 0.22 | 0.18 | 0.15 | 2.84 | 0.03 |
| Leaf chlorophyll (SPAD value) | 47 | 46.3 | 46 | 43.1 | 39.4 | 34 | 46 | 44.3 | 43 | 37.2 | 34.5 | 3.29 | 2.51 |
| **Variety: Neelem Desi** | | | | | | | | | | | | | |
| Germination (%) | 95 | 85 | 78 | 62.5 | 50 | 27.5 | 85 | 70 | 62.5 | 47.5 | 27.5 | 38.9 | 6.31 |
| Plant height (cm) | 54.1 | 53.9 | 51.2 | 47.7 | 43 | 36.9 | 52.6 | 48.6 | 43 | 39 | 36.2 | 9.54 | 2.78 |
| Leaf area (cm$^3$) | 69.9 | 65.6 | 60.8 | 56.4 | 51 | 49 | 64 | 60.2 | 55.2 | 50.4 | 48.2 | 8.03 | 2.83 |
| Leaf area (cm$^3$): 25d | 29.3 | 29 | 26.4 | 23.5 | 21 | 20 | 29.3 | 26.4 | 23.2 | 21.6 | 20 | 2.88 | 2.03 |
| Green leaf count | 8 | 8 | 7 | 7 | 7 | 7 | 8 | 7.5 | 7 | 7 | 7 | 1.98 | 0.17 |
| Fresh weight (g) | 36.6 | 36.4 | 35.7 | 33.4 | 29.5 | 25.9 | 36.6 | 35.5 | 32.6 | 28.8 | 24.8 | 2.11 | 1.94 |
| Dry weight (g) | 14 | 13.2 | 11.5 | 10 | 9 | 8 | 13 | 11 | 9.8 | 9 | 7.7 | 13.8 | 0.58 |
| Dry weight (g): 25d | 6.8 | 6.9 | 6.4 | 5.5 | 5 | 4.3 | 6.9 | 6.4 | 5.5 | 4.8 | 4.3 | 9.44 | 0.35 |
| RGR (g day$^{-1}$) | 0.48 | 0.43 | 0.35 | 0.3 | 0.26 | 0.24 | 0.42 | 0.32 | 0.29 | 0.26 | 0.21 | 3.52 | 0.02 |
| NAR | 0.18 | 0.16 | 0.15 | 0.14 | 0.14 | 0.13 | 0.15 | 0.14 | 0.14 | 0.13 | 0.11 | 3.73 | 0.01 |
| Leaf chlorophyll (SPAD value) | 46.1 | 46 | 45 | 41.4 | 37.3 | 34.7 | 44.6 | 42.5 | 39.7 | 35.7 | 33.2 | 4.19 | 1.97 |

**Notes.**
NS, statistically not significant.

where, $w_2$ = plant dw at harvest time of 40 d ($t_2$), $w_1$ = plant dw at harvest time 25d ($t_1$)

$$NAR = 2(w_2 - w_1)/(LA_1 + LA_2)(t_2 - t_1)$$

where, $w_2$ = leaf dw at harvest time 40 d ($t_2$), $w_1$ = leaf dw at harvest time 25 d ($t_1$), $LA_1$ = leaf area measured at harvest time 25 d ($t_1$), $LA_2$ = leaf area measured at harvest time 40 d ($t_2$).

## Determination of soil parameters

Soil texture (including clay 63%, sand 30%) was determined with the Bouyoucos hydrometer method by preparing a soil paste that was saturated with distilled water (*Sheldrick & Wang, 1993*). Soil pH (H$_2$O) and EC (mS/cm) were determined using a pH and EC meter, respectively. For this purpose, soil-water suspension 1:2.5 (w/v) was prepared, and the cathode of the meters was dipped into it (*Hassan et al., 2013c*).

### Determination of SOM

Soil organic matter content was determined by the method of *Walkley & Black (1934)*. For this purpose, reduction of Cr ion by soil organic matter and an unreduced $Cr_2O_7{}^{2-}$ was measured. A total of 0.5 g ground and sieved soil mixed with 10 mL $K_2Cr_2O_7$ (1M) followed by the addition of 20 mL conc. $H_2SO_4$. The sample was well shaken for 30 min and the final volume was raised to 200 mL by distilled water. Afterward, soil material was titrated against acidified 0.5 M ammonium ferrous sulfate. Reading of the sample was manipulated from blank upon the appearance of a green endpoint.

### Determination of soil urease activity

Soil urease activity (UA) was determined by the method of *Kandeler & Gerber (1988)* as described by *Hassan et al. (2013c)*. For this purpose, metal-treated 5 g soil was mixed with 10 mL of urea solution; and then 10 mL of buffer solution (citric acid, KOH, and NaOH) having pH 6.7 was also added to it. This solution was incubated at 37 °C for 24 h. After filtration, the solution was mixed with reagents (phenol + NaOH); to this solution, sodium hypochlorite solution was also added. The absorbance of the appeared blue color was noted at 578 nm through a spectrophotometer.

### Determination of soil respiration

For the determination of soil (microbial) respiration, a laboratory incubation experiment was performed to measure soil respiration under two different heavy metals following the method of *Anderson (1982)* as described by *Devi & Yadava (2009)*. Soil samples were moistened with the five respective doses of either metal and were placed in closed jars provided with test tubes containing NaOH and distilled water test tubes. Evolved carbon dioxide over time was trapped by NaOH titrated with the acid of known normality.

(formula: mg of $CO_2 = V \times N \times 22$)

where V = volume of acid used against 10 ml NaOH N = normality of acid used; and the value 22 is a factor for $CO_2$ evolved during reaction.

### Estimation of soil nutrients (N, P, K)

Soil $K^+$ was assessed by flame photometer taking a soil sample (2.5 g) by shaking with 33 mL of 1M KCl following *Anderson & Ingram (1993)*. The excess $K^+$ in the soil sample was washed three times with 95% ethanol and the adsorbed $K^+$ was then extracted by addition of 33 mL of 1M $NH_4OAc$. The volume of this extract was raised to 100 mL and further added with 1M $NH_4OAc$ to estimate $K^+$ in the extract. Similarly, soil phosphorus was assessed following the method described by *Olsen & Sommers (1982)*. For available phosphorus, 5 g of soil was obtained in a 250 mL flask, and 0.5M $NaHCO_3$ (100 mL) was added to it. This solution was shaken for 30 min and then the filtrate was collected. Ten mL of the filtrate was shifted into a flask of 50 mL along with 1 ml of 5N $H_2SO_4$ (sulfuric acid) and the volume was increased up to 40 mL by adding distilled water. To this solution, 8 mL of ascorbic acid as a reagent was added to develop color; and transmittance was recorded at 880 nm using a spectrophotometer. Soil nitrogen was determined by Kjeldahl's method

(*Ahmad et al., 2011*) using the formula:

$$N(\%) = \frac{\text{acid used for sample} - \text{acid used for blank} \times \text{acid normality}}{\text{volume of sample}} \times 14.01 \times 10 \times 100.$$

## Metal detection in soil and plants

For determination of soil metal content, maize plants were separated from contaminated pot soil was taken out from the respective pots. These samples were executed following the method of *Welz & Sperling (1999)* using an atomic absorption spectrometer (Perkin-Elmer). Hot acid digestion was used for a 1 g soil sample using 15 mL of the acid mixture in a 5:1:1 ratio (70% $HNO_3$, 70% $H_2SO_4$, and 65% $HClO_4$). After cooling, the transparent acidic solution was filtered (Whatman no. 42) and diluted with distilled water. Metal analysis was carried out at analytical spectral lines *i.e.,* Cr: 357.9 nm, and Pb: 283.3 nm. A similar digestion procedure was executed for plant metal detection. For metal accumulation and translocation study, different plant parts (root, stem, leaves, *etc.*) were used. In the treated plants, the bio-magnification ratio (BMR) and metal accumulation factor (ACF) was assessed by the method of *Baker et al. (1995)* whilst the metal translocation factor (TF) was calculated according to *Yanqun et al. (2005)* using the following equations:

$$BMR = PU/MA$$

$$ACF = PU/MT$$

$$TF = \text{Element (shoot)}/\text{Element (root)}$$

where, PU = metal concentration in whole plant ($\mu g\ g^{-1}$), MA = available metal concentration in soil ($\mu g\ g^{-1}$), MT = total metal concentration in soil ($\mu g\ g^{-1}$)

## Quality control analysis and assurance

Chemical analysis of samples was performed by AAS and spectrophotometer. High grade standard chemicals and glass ware were used (Merck-Germany). By using a calibration curve, calibration of instruments was executed with a series of standard solutions of varying concentrations. The chemical stock solution was prepared with double-deionized water. Glass ware was used after cleaning and rinsing with diluted $HNO_3$ to avoid some probable contamination. For quality results, each sample was analyzed in a repeated way by following the standard reference procedure (*Atta et al., 2023*).

## Data analysis

For comparison of the significance level of means under metal treatment, analysis of data was performed by calculating the F-value from ANOVA test using a statistical package IBM-SPSS (V. 20). While error graphs (LSD 5%) were prepared in MS-Excel.

# RESULTS

## Effect of Cr and Pb on plant growth-related parameters

### Seed germination

The Cr and Pb treatments suppressed the maize seed germination in a concentration-dependent manner (Table 1). The decrease in seed germination of both varieties was much more obvious at 150 ppm Cr and 100 ppm Pb treatment. At the highest concentration (300 ppm), Cr and Pb inhibited the seed germination of Pak-Afgoi by 67% and 64% and Neelem by 68% and 73%, respectively.

### Green leaves and leaf area

Table 1 shows a minor suppressive effect of Cr and Pb at the early growth stage of maize varieties. Green leaf count was not significantly affected in Pak-Afgoi as compared to the variety Neelem. Cr rapidly decreased this agronomic trait at 150–300 ppm (Pak-Afgoi 5–10%; Neelem 6–29%) and 100–300 ppm Pb application (Pak-Afgoi 5–10%; Neelem 18–29%). A decline in leaf count was more at 300 ppm of Cr and Pb. Likewise, leaf area was also decreased along with the increasing metal doses in a more significant way, and decreased much at elevated levels of Cr and Pb (300 ppm), whereas Neelem declined more than Pak-Afgoi (30 and 31%). Likewise, leaf area was also measured at 25th d (harvest-1) which decreased by 21–29% in Pak-Afgoi, while a decrease in Neelem was up to 32% under Cr and Pb stress.

### Plant height

Pb has more adverse effects on plant height than Cr. Comparatively, the Neelem variety showed a pronounced decreasing trend for this agronomic trait. During the early growth stage, the maximum plant length for Pak-Afgoi and Neelem was recorded up to 68 cm and 54.1 cm, respectively. Plant height decreased much at higher metal concentrations (300 ppm). Plant height decreased in Pak-Afgoi under Cr and Pb by 22% and 23%, respectively. Plant height also decreased in Neelem by 32% and 45%, respectively (Table 1). Both the varieties showed a tolerant behavior and were least affected at 50 and 30 ppm of Cr and Pb, whilst rapidly declining at 150 ppm Cr and 100 ppm Pb.

### Shoot girth

Both Cr and Pb affected the shoot girth of maize varieties in a declining and concentration-dependent- pattern. During the early growth stage, the maximum shoot girth of Pak-Afgoi and Neelem was 6.2 cm and 5.5 cm in the control treatment, whilst the mean decrease in shoot girth of Pak-Afgoi *viz.* varying concentration of Cr and Pb metals was 11-11.3% and 13%, respectively. However, at 300 ppm of Cr and Pb; the mean decrease was more than 19% (Pak-Afgoi) and 27% (Neelem). Maize variety Neelem was less affected than Pak-Afgoi for this circumference trait (Table 1).

### Plant biomass

Plant fresh weight and dry weight were assessed under two different metals that revealed metal-induced toxicity on plants. Statistical analysis has predicted a less significant effect of metals on the fresh weight of Pak-Afgoi than of Neelem. Comparative to the control value

(46 g) and to the low concentrations of Cr and Pb, fresh weight decreased more up to 17% at 300 ppm. Fresh weight of Neelem in the control treatment was 26.6 g later decreased up to 27% by Cr and 32% by Pb, likewise Pak-Afgoi (Table 1).

Plant dry weight also decreased both at harvest-1 and harvest-2 (Table 1). Pak-Afgoi attained a maximum of 16.2 g dry weight that decreased up to 38% (Cr, Pb). Similarly, Neelem attained 14g dry weight as the control value and later underwent a significant decrease due to metal toxicity at 300 ppm of Cr (43%) and Pb (45%). Overall, data shows a rapid decrease in plant biomass parameters observed from 150 ppm (Cr) and 100 ppm (Pb) than at lower metal concentrations.

### Relative growth rate (RGR) and net assimilation rate (NAR)

Data about RGR (g d$^{-1}$ increase in dry matter) revealed that both maize varieties continued growing *viz*. control and metal treatments and gained dry matter between the two harvests (H2-H1). However, a mean decrease in plant growth rate was noted by 28% & 31% (Pak-Afgoi), 31% & 37% (Neelem) under Cr and Pb, respectively. At maximum dose (300 ppm) of Cr and Pb, RGR decreased by 48% & 49% in Pak-Afgoi, while 53% & 56% in Neelem, respectively.

The decreasing trend of RGR and dry matter of plants strongly showed an affected accumulation of metabolites/photosynthate between the two harvests due to metal stress. A similar decreasing trend was observed in the case of NAR. Metal treatment has revealed increased metal toxicity from H1-H2 (as the plant spent more under stressful conditions). A mean decrease in NAR under the Cr effect in Pak-Afgoi and Neelem was 10% & 17%, whilst Pb affected this parameter by 21% & 22.2%, respectively. The decline in RGR and NAR was elevated at elevated metal concentrations (Table 1).

## Effect of Cr and Pb on soil physicochemical properties
### Soil pH

Soil pH of the control soil was 7.8, lowered up to 6.7 and 6.5 under Cr and Pb application at H2 (day 40), respectively. Results showed a mild effect of metals on soil pH up to 100 ppm Cr and 60 ppm Pb during this incubation period. The effect of metals increased and lowered soil pH more at elevated metal doses (Table 2).

### SOM content and urease activity

The control value of SOM at H2 was 2.8%, whilst a rapid decrease in SOM content was initiated at 150 ppm Cr and 100 ppm Pb. The mean cumulative decrease in SOM content was 17% & 20% under Cr (50–300 ppm) and Pb (30–300 ppm) toxicity. At 300 ppm Cr and Pb application, SOM content decreased by 43% and 46%, respectively (Table 2). Soil urease activity (UA) was found affected by different concentrations of the metals. At harvest time (day 40), the mean decrease in UA was up to 22% & 26% due to Cr and Pb, respectively. Table 2 shows that enzyme activity was much more affected at the elevated metal concentrations than at lower. Although lower metal doses had the least effect on UA, a clear decline in UA was initiated at 100 ppm level of the metals which turned to its peak at 300 ppm *i.e.,* decreased under Cr (41%) and Pb (47%).

Peer J

**Table 2  Assessment of some soil parameters at harvest-2 (40th day) under Cr & Pb stress.**

| SOV | Cr (ppm) | | | | | | Pb (ppm) | | | | | F-value | LSD (5%) |
|---|---|---|---|---|---|---|---|---|---|---|---|---|---|
| | **0** | **50** | **100** | **150** | **250** | **300** | **30** | **60** | **100** | **150** | **300** | | |
| pH | 7.6 | 7.4 | 7.4 | 7.2 | 6.6 | 6.4 | 7.2 | 7 | 6.7 | 6.4 | 6.3 | 2.5 | 0.1 |
| SOM (%) | 2.8 | 2.6 | 2.46 | 2.31 | 2 | 1.8 | 2.7 | 2.51 | 2.36 | 2 | 1.8 | 19 | 0.14 |
| Urease Activity (mg NH4-N kg$^{-1}$ 24 h$^{-1}$) | 14.6 | 14 | 13.2 | 11.1 | 9.7 | 8.5 | 14.2 | 12.5 | 10.7 | 9.2 | 7.6 | 15.3 | 1.01 |
| CO$_2$ evolution (mg): | 176 | 171 | 165 | 153.4 | 145 | 124 | 177 | 171.3 | 160.7 | 149.1 | 135 | 12.7 | 6.54 |
| **Soil nutrients** | | | | | | | | | | | | | |
| N (%) | 2.82 | 2.5 | 2.3 | 2.1 | 1.9 | 1.62 | 2.4 | 2.2 | 2 | 1.6 | 1.45 | 22.1 | 0.47 |
| P (ppm) | 8.2 | 8.2 | 8 | 7.7 | 7.3 | 6.6 | 8.1 | 8 | 7.7 | 7.3 | 6.3 | 2.9 | 0.18 |
| K (ppm) | 182 | 178 | 172.4 | 155 | 142.1 | 124.1 | 179.2 | 177 | 160.2 | 143 | 128.6 | 9.6 | 9.03 |

### Soil respiration: (evolution of $CO_2$)

Table 2 shows the impact of soil metals on the amount of soil $CO_2$ under cultivation of maize varieties. This parameter of the soil decreased from 176 mg to 124 mg. The mean effect of metal treatments (Cr 50–300 ppm; Pb 30–300 ppm) showed metal toxicity following a significant decrease in soil respiration (SRP), compared to control. SRP decreased more by Cr (14%) than Pb (9.5%) showing much Cr-toxicity on SRP. However, differentiating the treatment effect, 300 ppm Cr and Pb level imposed drastic effects on this parameter by 29% & 23%, respectively.

### Soil nutrients (NPK)

Both Cr and Pb treatments have toxic effects on available soil nutrients. Soil nitrogen (N), phosphorus (P), and potassium (K) contents were decreased gradually at lower metal concentrations than at maximum levels. Comparing the toxicity of Cr and Pb on NPK contents at harvest (day 40), the mean decrease in N, and P was more under Pb stress (31%, 7%) than Cr stress (25%, 7%). Likewise, $K^+$ was less affected under Pb stress (13%) than Cr (15%), respectively. The decreasing order of NPK at the maximum metal concentration of Cr and Pb was 42%, 17%, 32%, 47%, 22%, and 29%, respectively (Table 2).

### Metal uptake from soil (bioaccumulation in plant tissues)

At harvest-2, plant parameters showed a substantial effect of metal toxicity along with the increasing metal concentrations. Cr accumulation was more in the stem than roots and leaves, whilst Pb accumulated more in the roots than stems and leaves of the test varieties (Fig. 1). In Pak-Afgoi and Neelem, maximum Cr was accumulated in the stem (61.2 µg/g & 68.5 µg/g) at its highest concentration in the soil medium. Similarly, Pb accumulation in the stem part of Pak-Afgoi and Neelem was observed at 17.2 µg/g and 19.2 µg/g, respectively. However, Pb contents in the roots were 47.3 µg/g in Pak-Afgoi and 51.2 µg/g in Neelem. Although, metal accumulation in different plant parts was increasing way, however, a hasty metal up taken by plants occurred at 150 ppm (Cr) and 100 ppm of Pb (Fig. 2).

The phytoremediation potential of maize varieties was assessed by calculating the metal accumulation factor (ACF), translocation factor (TF), and bio-magnification ratio (BMR). Results clearly showed significantly increased values of Cr and Pb metals for ACF, TF, and BMR by variety Neelem than Pak-Afgoi. However, both the varieties significantly removed Cr and Pb content from the soil and accumulated in different parts successfully (Fig. 3).

## DISCUSSION

Results from the Petri plate experiment have shown that both the tested varieties undergo abiotic metal stress and seed germination decreased with the increasing concentration of Cr and Pb. Studies on seed germination characteristics showed its inhibition under metal (Pb) toxicity even at low or micro-molar levels (*Kopittke, Asher & Menzies, 2008*). However, there are few reports about the progression of seed germination and inhibition of radical/ hypocotyl length in *Elsholtzia argyi* (*Islam et al., 2008*), but the same was not observed in the present course of the investigation. All the concentrations of Cr and Pb were found

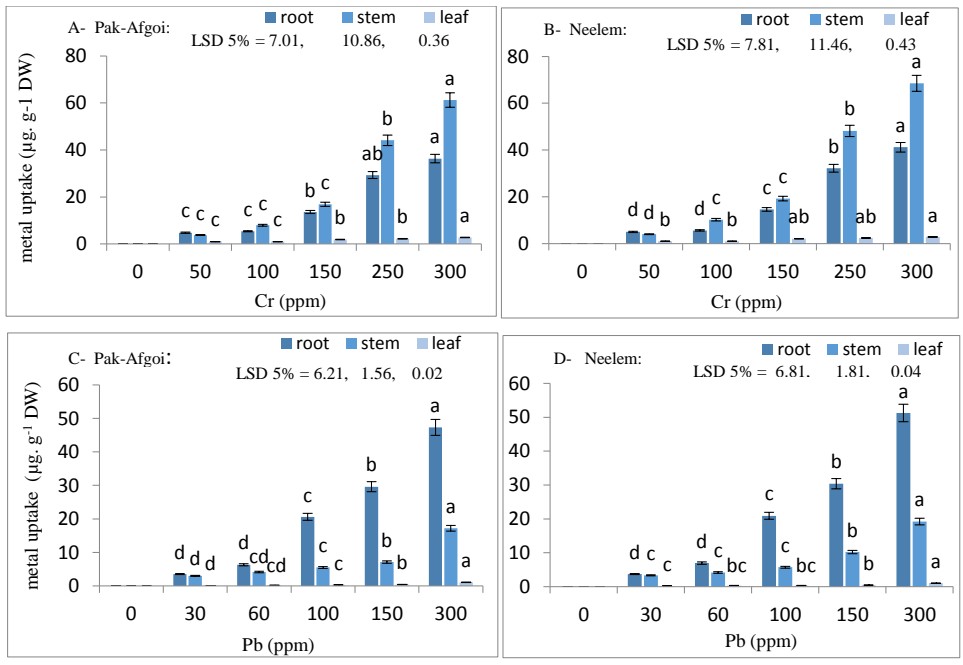

**Figure 1  Metal partitioning ($\mu$g g$^{-1}$) in different plant parts of Pak-Afgoi & Neelem maize at 40 d (A–D).**

to be inhibitory for maize germination and, inhibition exceeded along with the increasing metal concentrations in the medium. Arguments by *Sengar et al. (2008)* revealed inhibition of germination due to interference of Cr and Pb with the essential enzymes for seed germination (amylase and protease). Moreover, *Atici, Agar & Battal (2005)* documented that inhibition of GA$_3$ (gibberellic acid) and activation of ABA (abscisic acid) during germination of *Cicer arietinum* (chickpea) were due to Zn, Pb, and Cd metals.

Heavy metals are considered the major environmental toxins that adversely affect all living organisms including plants (*Ashraf et al., 2018*; *Bargagli et al., 2019*). The toxic effects of metals on different growth attributes in plants are due to abnormal nutrient uptake from plant roots as metals become stuck in roots and oppose nutrient uptake from the soil (*Singh et al., 2016*). Different agronomic parameters of rice plants *i.e.,* plant length, tiller count, and dry weight biomass undergoes significant reduction due to Pb doses 0.6 mM–1.2 mM. Observations highlighted less toxicity at a lower Pb dose of 0.6 mM than at 1.2 mM of Pb. A decline in the length of rice plants at the maximum Pb dose was 13% and dry weight decreased by 61% in cultivar Ilmi (*Khan et al., 2021*). A similar observation was reported by *Orhue & Ekhomun (2010)*. Cr affected plant height and dry matter in waterleaf after 100 mg Cr dose. Reduced plant length was due to Cr accumulation that suppressed mitotic activity in the affected plants. Decreases in plant fresh weight due to Cr-toxicity at the varying extent of Cr concentrations also have been reported in *Hibiscus esculentus* (*Amin et al., 2013*), *Helianthus annus* (*Fozia et al., 2008*), and *Brassica oleracea* (*Ozdener et al., 2011*). These studies also revealed that a decrease in growth and biomass parameters

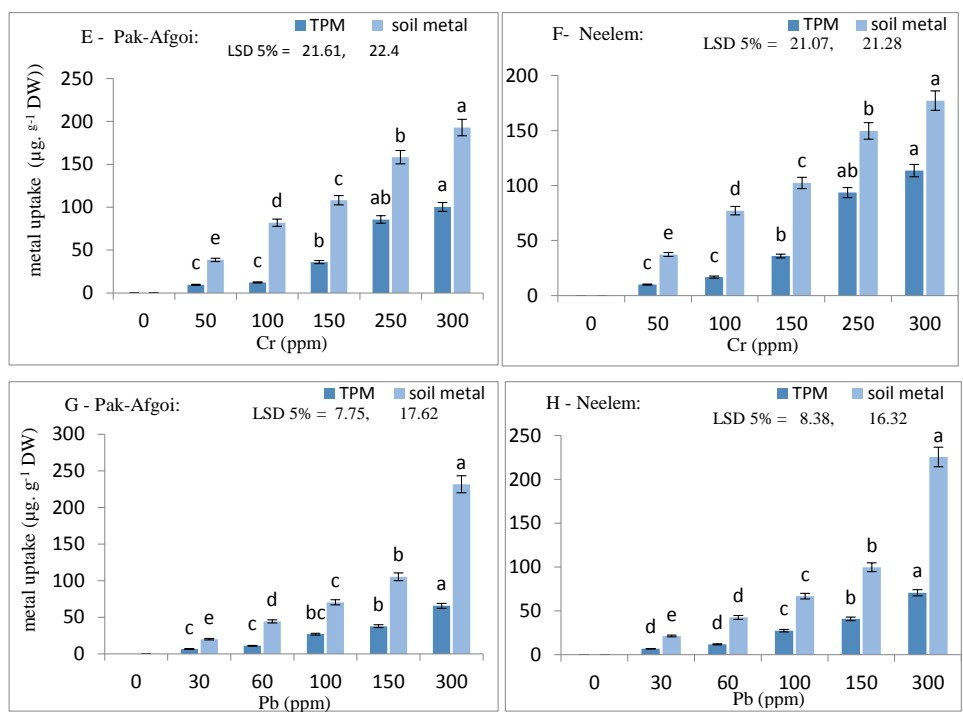

**Figure 2** Total metal uptake by plant (TPM) and soil metal retention ($\mu g\ g^{-1}$) at 40 d (E–H).

of the subsequent plants occurs due to increasing metal levels in the growth medium. In agreement with these earlier studies, the present study has also revealed the negative effect of increasing concentrations of Cr and Pb on maize growth (Fig. 4).

Plant growth is attributed mainly to leaf characteristics and the photosynthetic performance of plants. Metals (Cr and Pb) are well-known abiotic stressors that inhibit the photosynthetic performance of intact plants and finally affect plant growth and biomass yield (*Houri et al., 2020*). Altered values of RGR also predicted the affected plant growth whereas, NAR revealed the negative impact of metals on photosynthetic performance and product of photosynthesis *i.e.,* dry matter content. A leaf is an important photosynthetic organ of plants that plays a key role in the growth of plants. Pb and Cr adversely affect the growth and development of leaves in *Lycopersicon esculantum*, *Pisum sativum,* and *Zea mays* (*Yoon et al., 2006*; *Anjum et al., 2016*). Studies showed the inhibitory role of heavy metals on leaf growth and development in rice plants through the generation of oxidative stress/ROS (*Singh et al., 2020*). These studies are strappingly evident the findings of the present study that leaf number and leaf area in tested plants of maize significantly declined upon exposure to Cr and Pb doses.

Chlorophyll is one of the crucial molecules to facilitate photosynthetic activity in plants and is responsible for the electron transport chain to step forward the photosynthesis. However, heavy metals are responsible for altering the chloroplast structure and cause inhibition of the electron transport system by affecting its biosynthesis (*Wakeel, Xu & Gan, 2020*) through increased activity of chlorophyllase reported reduced chlorophyll

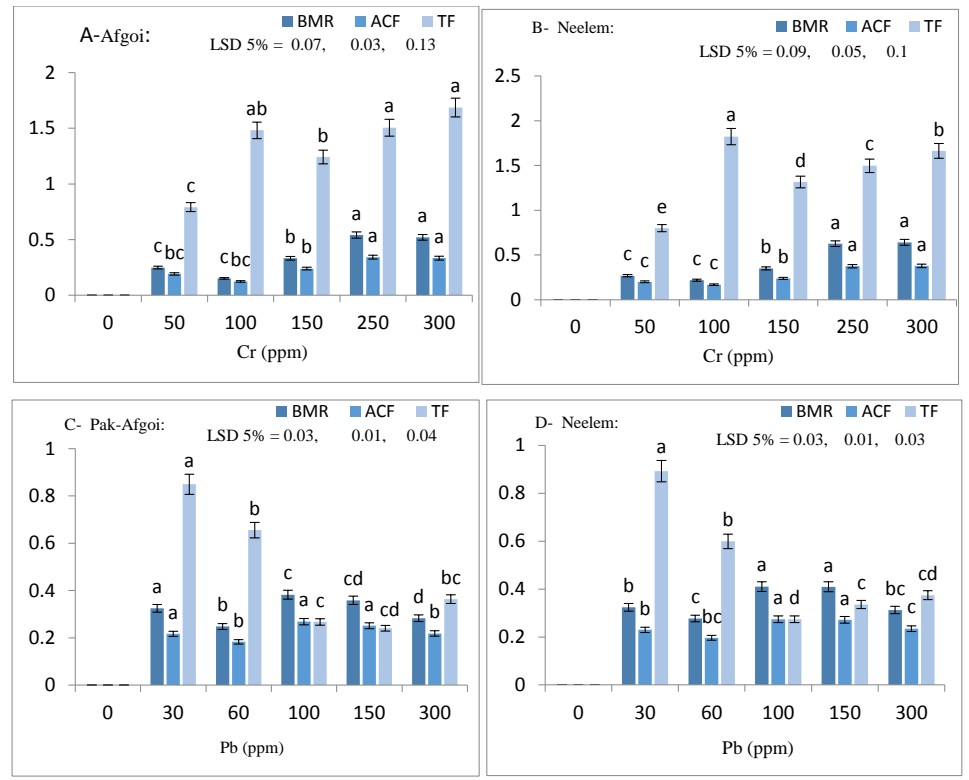

**Figure 3** Metal bio-magnification ratio (BMR), metal accumulation factor (ACF) & translocation factor (TF) at day 40 in Pak-Afgoi & Neelem maize (A–D).

biosynthesis and affected activity of NADPH protochlorophyllide oxidoreductase enzymes under Hg-toxicity. In continuation, *Singh et al. (2020)* investigated $Cr^{+6}$-induced alterations up to 89% in the chlorophyll content of mung beans (*Vigna radiata* L). A greater decline was at the highest concentration of 120 μM than at lower doses of 60–90 μM. Studies on Cr and Pb stress in *Nicotiana tabacum* and *Cicer arietinum* by *Bukhari et al. (2016)* and *Singh et al. (2020)* have also supported the findings of the present study. Heavy metals cause land degradation through soil acidification that happens due to the leaching mechanism of toxic metal ions. Soils presenting a low pH profile make metals to be available for growing plants and thus reduce crop yield (*Xu et al., 2012*). Although soils pose resistance to the pH change and act as a buffer (*Curtin & Trolove, 2013*); the long-term application of heavy metals put acidification effects on the subsequent soils. Heavy metals undergo hydrolysis in a solution of such soils, generate H $^+$ ions, and lower the pH (*Motuzova, Makarichev & Petrov, 2011*; *Schwertfeger & Hendershot, 2012*). Consequently, soil acidification results in nutrient depletion and affects crop plants (*Najafi & Jalali, 2016*).

Soil organic matter (SOM) is assumed as a potential source for microbial activity in the agricultural soils and releases nutrients into the soils through the degradation of soil organic components. Likewise, SOM also shows a large sorption affinity toward metals (*Yin et al., 2002*). Likewise, microbes release certain enzymes of key value into the soil.

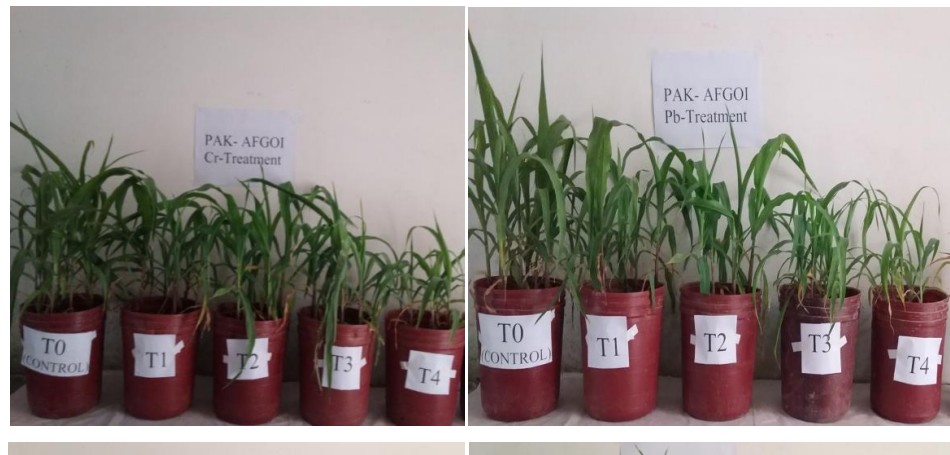
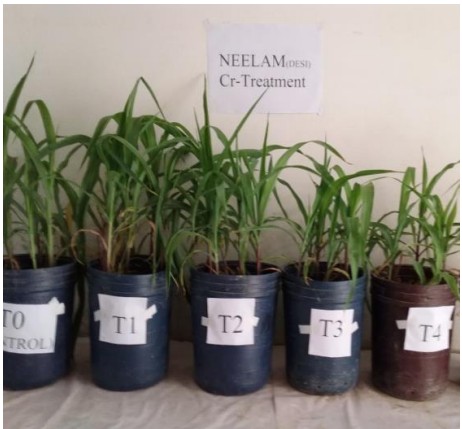
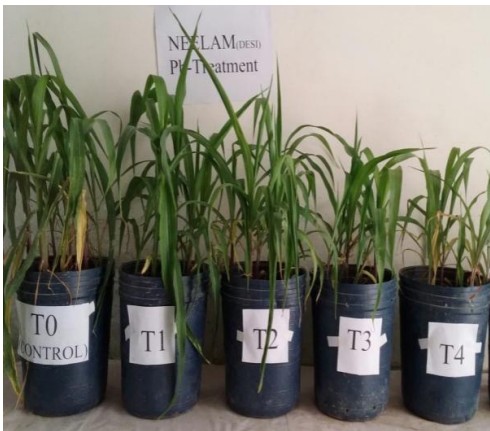

**Figure 4** Comparison of growth/height of trialed maize plants from two varieties PakAfgoi & Neelem under Cr & Pb treatment.

These enzymes (urease, dehydrogenase, and phosphatase) are dynamic in the biochemical functionality of the soils including the decomposition of SOM. Hence, the soil enzymes are referred to as an indicator of soil quality, a good measure of soil microbial activity with the recycling of nutrients from the decomposed SOM (*Puglisi et al., 2006*). Soil urease is a sensitive hydrolyzing enzyme and is a potential indicator of soil pollution and biological activities (*Hinojosa, Carreira & García-Rułz, 2004*). Moreover, despite the positive role of urease in soil chemistry; the addition of varying doses of Cu, Cd, Zn, Pb, and Cr decreases the activity of soil urease at large, in contrast depending on the incubation period (*Malley, Nair & Ho, 2005*; *Shen et al., 2005*). This might be correlated with the decomposition of SOM.

Soil respiration ($CO_2$ evolution) is another parameter to assess soil microbial performance with the decomposition of SOM in the subsequent soils (*Nawaz et al., 2015*). *Verma et al. (2010)* have reported a decreased rate of SOM decomposition under Cd, $Cr^{+6,}$ and Pb stress vide different incubation periods. Toxicity of all three metals was found to increase towards SOM-decomposition and soil respiration ($CO_2$ evolution) along with the increasing metal treatment and incubation period. Investigations by *Algaidi (2013)*

have extended the Zn, Pb toxicity on aerobic bacteria. An elevated level of both metals significantly decreased the physiological activities of soil microbes and $CO_2$ production. Similar evidence has been provided by *Mathe-Gaspar et al. (2005)* for Zn, Cu, and Cd metal ions. Soil microbial activity and biodegradation of SOM content play a vital role in the soil fertilization process, cycling of nutrients, and hence increasing the soil fertility (*Kumar et al., 2019*). Likewise, the soil urease enzyme has a potential role in soil N-cycle due to its hydrolytic properties. Consequently, the difference between pre and post-harvest soils revealed a remarkable decline in the available nitrogen under Cr and Pb stress is in agreement with the experimental outcomes of *Orhue & Ekhomun (2010)* *i.e.,* increasing concentration of $Cr^{+6}$ continues to decline the soil N by 39% at highest Cr dose 200 mg. A similar observation was reported for soil P and K availability in the present study, indicating the Cd-affected activity of dehydrogenase and phosphatase (*Hassan et al., 2013c*). The present study also revealed similar effects of Cr and Pb on soil macronutrients and indicated the metal toxicity on mineral cycling with the affected SOM and enzymatic activity *i.e.,* urease for N-cycling in the treated soil.

Crops grown on metal-contaminated soils have a greater accumulation of these metals than crops grown in uncontaminated soil (*Sharma, Agrawal & Marshall, 2008*). Plants have a natural capacity to absorb metal ions from the soil even in low concentrations through their root system. To attain efficient reclamation of metal-contaminated soil, plant roots form a rhizosphere ecosystem, absorb and accumulate the heavy metals and improve soil fertility (*Jacob et al., 2018*). Hyper-accumulator plant species were found to be effective in the removal of metals. Plant species that have the potential of accumulating a major portion of metals from the soil are referred to as hyper-accumulators; and are used in phytoremediation techniques to remove the pollutants (*Clemens, 2006*). However, phytoremediation potential exactly depends on the plant's capacity to extract heavy metals from the intact environment and bio-accumulating them in various plant parts without having adverse effects on soil structure, fertility, and biological activity (*Yan et al., 2020*). For instance, *Paspalum fasciculatum* showed the potential of accumulating Cd and Pb in declining order of metal concentrations in roots >leaves >stem. Cd uptake was recorded more than Pb, revealing this plant to be phytostabilizing as the maximum Cd amount was accumulated in roots (*Salas-Moreno & Marrugo-Negrete, 2020*). Likewise, phytostabilization of ryegrass (*Lolium perenne* L.) also showed to be potential for removal of Cd, Cu, As, Zn, and Cr (*Alvarenga et al., 2009*). *Panicum* grass also exhibited a maximum accumulation of Pb in roots than in shoot *i.e.,* roots accumulate 96% more lead as compared to shoot (*Pires-Lira et al., 2020*). In the present study, uptake of Cr and Pb (Afgoi 33%, 22%; Neelem 37%, 24%) at 300 ppm by maize varieties during EGS has uncovered the emergent potential *i.e.,* hyper-accumulation and phytostabilization of this crop cultivated under particulars soil and environmental conditions of Dera Ghazi Khan. A higher portion of Cr metal was observed in stem tissues than in roots and leaves. Likewise, Pb accumulation was more prominent in roots than stems and leaves of the subsequent maize plants. The phytoaccumulation potential of maize variety Neelem was more remarkable than Pak-Afgoi, indicating maize varieties as hyper-accumulators for Cr with phytostabilization for Pb metal. These maize plants contain higher Cr and Pb levels than the WHO/FAO

permissible limits *i.e.,* Cr 2.3 mg/kg and Pb: 0.3 mg/kg (*Adu, Aderinola & Kusemiju, 2012*). The contaminated maize plants are recommended as unsafe for health and be destroyed systematically by burning in high-temperature cement kiln of *D.G. Khan Cement Company* which is available in the study area. Moreover, in the future, screening of different native plant species for phytoremediation purposes along with the focus on their biochemical responses, and tolerance mechanisms is suggested. Application of phosphorus increases soil fertility through increased microbial activity and improves soil nutrient status. The efficacy of soil enzymes (urease, phosphatase) to recycle the nutrients turns high due to phosphorus implication in the contaminated soil (*Iqbal et al., 2023*). Similarly, the microbial role of bioremediation may be another choice to reclaim contaminated soils. At present, to prevent further addition of heavy metals into agro-soils, irrigation with municipal and industrial wastewater should be banned or if irrigation with the wastewater is continued, it should be recycled through wastewater treatment plants (*Atta et al., 2023*).

## CONCLUSION

Both heavy metals are toxic to seed germination, plant height, leaf development, plant biomass, and chlorophyll content. Moreover, RGR and NAR values of both varieties also indicated the suppressive role of Cr and Pb in the soil medium. The affected plant growth mechanism showed its affinity with the affected soil characteristics under metal stress, which was much more on higher metal levels. Data has revealed an acidic effect of heavy metals on the soil and affected soil respiration by affecting soil microbial activity. Likewise, decreasing levels of soil enzymes have revealed an affected decomposition of SOM content and recycling of soil nutrients. However, besides the toxic effect of metals, maize plants showed great potential in accumulating/partitioning Cr and Pb from the subsequent rhizospheric pot soils. Therefore, it is suggested that maize test varieties (Pak-Afgoi & Neelem Desi) be grown as a tool of phytoremediation in the contaminated agro-soils of Dera Ghazi Khan District. However, such contaminated maize plants are recommended unsafe, and carcinogenic to use due to exceeding amounts of Cr and Pb metals than permissible limits, and be destroyed through cement kiln burning.

## ACKNOWLEDGEMENTS

Present work is a part of the Ph. D research of Muhammad Imran Atta. The authors are very grateful to Dr. Muhammad Khuram Afzal, Associate Department of Food Science (Bahauddin Zakariya University, Punjab-Pakistan) for his valuable suggestions and inputs. The authors also acknowledge the significant input and suggestions of Mr. Muhammad Kaleem (Principal Scientific Assistant, PAEC-LAB).

### Funding

This work was supported by Project number (RSP2023R298), King Saud University, Riyadh, Saudi Arabia. The funders had no role in study design, data collection and analysis, decision to publish, or preparation of the manuscript.

### Grant Disclosures

The following grant information was disclosed by the authors:
King Saud University, Riyadh, Saudi Arabia: RSP2023R298.

### Competing Interests

Habib Ali is an Academic Editor for PeerJ.

### Author Contributions

- Muhammad Imran Atta conceived and designed the experiments, performed the experiments, prepared figures and/or tables, and approved the final draft.
- Syeda Sadaf Zehra conceived and designed the experiments, performed the experiments, prepared figures and/or tables, and approved the final draft.
- Habib Ali analyzed the data, prepared figures and/or tables, and approved the final draft.
- Basharat Ali analyzed the data, prepared figures and/or tables, and approved the final draft.
- Syed Naveed Abbas conceived and designed the experiments, authored or reviewed drafts of the article, and approved the final draft.
- Sara Aimen conceived and designed the experiments, authored or reviewed drafts of the article, and approved the final draft.
- Sadia Sarwar conceived and designed the experiments, authored or reviewed drafts of the article, and approved the final draft.
- Ijaz Ahmad conceived and designed the experiments, authored or reviewed drafts of the article, and approved the final draft.
- Mumtaz Hussain conceived and designed the experiments, authored or reviewed drafts of the article, and approved the final draft.
- Ibrahim Al-Ashkar analyzed the data, prepared figures and/or tables, and approved the final draft.
- Dinakaran Elango analyzed the data, prepared figures and/or tables, and approved the final draft.
- Ayman El Sabagh analyzed the data, prepared figures and/or tables, and approved the final draft.

### Data Availability

The raw measurements are available in the Supplementary File.

## Supplemental Information

Supplemental information for this article can be found online at http://dx.doi.org/10.7717/peerj.16067#supplemental-information.

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
