# Peer review of "Assessing the effect of heavy metals on maize (Zea mays L.) growth and soil characteristics: plants-implications for phytoremediation"

_PeerJ, doi:10.7717/peerj.16067_

## Round 0.1 · original submission · Major Revisions

Please submit the revised manuscript after incorporating the comments of all reviewers and a point-to-point rebuttal letter. Moreover, please highlight the literature gap and the contribution of the study in the introduction section. The language of the paper also needs attention.
Reviewer 1 ·

Basic reporting

Dear editor,
This manuscript presents important findings that can contribute to crop and soil management under heavy metal pollution that is negatively affecting soil health and agricultural productivity. I suggest accepting the article for publication in Peer J, if the authors further improve the manuscript in certain areas with moderate modification. Here are some of the issues in the article along with my suggestions.
Title: Although, presentation of this research is graded good. However, the title of manuscript is not meeting the “ criteria of manuscript reflection” and is suggested to be revised carefully.

Abstract/ Background, Methods, Results: This section needs a technical revision. Brief description of methods used and presentation of results summary is insufficient. Here, role of heavy metals for soil properties has not been added.
Introduction
Provide context for the study: In the introduction, it would be helpful to provide a brief overview of previous studies or literature on the topic to establish the existing knowledge and the research gap that this study intends to address. This will help readers understand the significance and novelty of the research.
Review the text to ensure that grammar and punctuation are correct for improved readability.
L35: Check spellings of remdiation.
L37: Chromium…….regarded…. this word regarded should be replaced with an appropriate word like ranked etc.
L39: Cr is used…., here authors used an abbreviation to represent a specific metal. My suggestion is to place the abbreviation (Cr) after the word chromium in the start of this para. It would be an easy understanding for a reader.
L42: Has a plant (any) posses a single mechanism to uptake metals? This sentence should be revised.
L49-50: sentence (s) should be revised ( removal of unnecessary use of fullstop)
L61: Word criterion should be replaced with parameter
Materials and methods
This section needs addition of quality control and assurance
L108: Mention chemical source of metals.
*Generally readout this section and simplify the text. Remove unnecessary commas etc.
Results and Conclusion
In overall, this section looks well explanatory. However I would like some suggestions as under:
Provide more specific details: In certain instances, additional details could be provided to support the statements made. For example, when mentioning previous studies, it would be helpful to provide the specific findings or results from those studies to strengthen the argument.
L225: Please write proper word…and… instead of symbolic … &…also do not repeatedly indicate “significant toxic effect of heavy metals (Cr, and Pb)”, instead write like “significant toxic effect of Cr, and Pb”. Please apply this change to whole manuscript especially the Results and Discussion sections.
L227: Pb puts instead of put
L238: Cr & Pb: please replace with Cr and Pb
L242: Plant biomass is sufficient in the main heading. Use fractions of biomass in the next line as it is in L243. Remove plant fresh wight/dry weight in L242.
L256: is confusing please rephrase
L257: again use of Cr & Pb…..
L272: please rephrase this line
L315: Avoid repeatedly use of “significantly”….
L319: outcomes of…….change it as… seed germination….
L244-345: rephrase the text here.
L349: remove as compared to control….. because altered values are taken/assessed when compared with the metal affected plants
L364: add scientific/botanical name of mung-bean. Please also made similar changes in the remaining text.
L407-408: please rephrase the text and simplify it
L411: How maize plant is ranked as hyperaccumulator…?
L447: What is the understanding of “predicted”…. Can you change or replace it with some other suitable word

Experimental design

no

Validity of the findings

no

Additional comments

no

·

Basic reporting

The paper on Phytoremediation Potentials of Maize Varieties Under Varied Levels of Chromium and Lead
is written in good order. It is according to the scientific standards and will attract the researchers who are working on influence of heavy metals on various field crops and soil properties. But before going to publications, there are some modifications required to make it more accurate.
• Before going deep into the manuscript, title of the manuscript is not looking up to the mark. I have some suggestions for the authors to modify title like this:
1. Assessing the effect of heavy metals on maize (Zea mays L.) growth and soil characteristics: plants-implications for phytoremediation
• Abstract section includes research background, methods and results. This part of the manuscript has some deficiencies……. Please read it deeply, and add some more appropriate description. In my opinion, methods also need the effect of Cr and Pb on the soil parameters etc. (Line 19-33)
• Similarly, keywords also need some addition as the authors should add soil parameters (Line 34-35)
• Please remove “paid a lot of attention” from the sentence…(Line 70)
• Please replace word “likewise” with “similarly” (Line 71)
• Some efficient…….. revive the metal affected agro-soils…..(Line 75). The word revive should be replaced with Restore.
• Authors are discussing the role of heavy metals in the soil medium…. Here in line 78, the word pollutants is used? Appropriate pollutants (heavy metals) should be placed.
• If phytoremediation is the technique / use of plants to remove heavy metals from a soil then why you have used additional/unnecessary sentence….(from plant body: Line 80). Author is advised to remove it.
• Therefore, a pot experiment was set……Line 98). It should be …..a pot experiment was conducted….
• Line 108: stock solution: where is the source agent/company of the used chemicals?
• Line 153: What is meant by to this (after phenol + NaOH)….. it should be …to this solution…
• volume of this extract was made up to 100 ml by further addition of 1M NH4OAc (Line 173). Here, in this sentence replace made with Raised….. and further addition of with by adding
• What is the understanding of averaged sampling? Line 188 remove word averaged.
• Apply ‘’used’’ instead of implemented
• What is understanding of subsequent plants, replace it with treated plants(Line 195)
• What is EGS? Please avoid abbreviation (Line 236)
• By contrasting with control and treatment effect….(Line 239-240). Effect of metal treatment is always understood for comparison with the control. So, in my opinion, remove contrasting with control.
• Remove …plant fresh and dry weight from plant biomass (line 242), as both are the components biomass and these fractions are defined in the next line 241.
• Cr affected mean NAR in Pak-Afgoi and Neelem by 10.34% & 17% (line 266-267). This sentence is just unclear what authors are asking/defining for? It should be like this: In overall, Cr affected NAR by 10.34% & 17% in Pak-Afgoi and Neelem maize.
• Statistical analysis (F-values & LSD)… there is no need to discuss statiscal analysis here (Line 273). It should be as “ Table 2 has showed… as the statistical values are presented in this table.
• by the subsequent concentrations of the applied metals (Line 280-81). Please write as “ by different concentrations of the metals”.
• By comparing the control, mean values of treatments (Cr 50-300 ppm; Pb 30-300 ppm) showed a remarkable decrease in soil respiration (Line 288-89). Modify this line as “ Mean effect of metal treatments (Cr 50-300 ppm; Pb 30-300ppm) showed metal toxicity following significant decrease in soil respiration, compared to control”.
• Line 294: metal treatments… please modify with “both Cr and Pb treatments”
• Line 296: with NPK add name of the nutrient in brackets
• Undergo metal’s stress be replaced with abiotic metal stress (318)
• Seed germination traits…. Here replace word traits by seed germination characteristics (320)
• ….growth attributes in plants could be due to abnormal nutrient uptake from plant roots (Line 332). It looks authors are confused with the previous study by Singh et al., 2016. The words “could be due to” should be replaced with “are due to”…..
• Different agronomic characteristics of rice plants should be as “ different agronomic parameters of rice plants” (Line 334). Also replace remarkable with significant (L 334).
• Line 354: please remove right hand small bracket ) placed after role of heavy metals. It is confusing the sense of the sentence.
• Chlorophyll is one of the crucial molecules to facilitate photosynthetic activity in plants, and is responsible for the electron transport chain to step forward this phenomenon (Line 358-359). This sentence becomes confusing when you are discussing role of chlorophyll in photosynthesis due to use of words “this phenomenon”. Simply replace it with “photosynthesis”.
• Line 388: What is the understanding of soil respiration at the start of para? While In the upcoming lines of the same para, authors also referred it as CO2 evolution. It would be more appropriate that if authors add CO2 evolution after soil respiration at very early start of para.
• Line 435-436: Authors repeatedly used moreover that looks in-appropriate. It is suggested to replace moreover (Line 436) with similarly.
• Papers citations: Additionally, I recommend some new publications list which the author can incorporate in the revision which are as follows:
1. The effect of soil microplastics on the Oryza sativa L. root growth traits under alien plant invasion. Frontiers in Ecology and Evolution. 11, 1172093.
2. Advancing Agro-Ecological Sustainability through Emerging Genetic Approaches in Crop Improvement for Plants. Functional and Integrative Genomics. 23, 145. https://doi.org/10.1007/s10142-023-01074-4
3. The high phosphorus incorporation promotes the soil enzymatic activity, nutritional status, and biomass of the crop. Polish Journal of Environmental Studies 32 (3), 158765. https://doi.org/10.15244/pjoes/158765
4. Combined inhibitory effect of Canada goldenrod invasion and soil microplastics on rice growth. International Journal of Environmental Research and Public Health. 19 (19), 11947.
5. Impacts of soil microplastics on the crops: A review. Applied Soil Ecology. 181, 104680.
• Conclusion: This section found good with seldom need of modification.
• Overall I suggested for minor revision of this manuscript

Experimental design

No comments

Validity of the findings

No comments

Additional comments

No comments

Reviewer 3 ·

Basic reporting

The manuscript is clear and professional English used throughout.

Experimental design

Methods described with sufficient detail & information to replicate.

Validity of the findings

Conclusions are well stated, linked to original research question & limited to supporting results.

Additional comments

In the present study “Phytoremediation Potentials of Maize Varieties Under Varied Levels of Chromium and Lead”, authors concluded that maize has phytoremediation potential to combat the heavy metal issue in the future under the particular environmental conditions of the area. I think that the work falls into the scope of the journal and findings are interesting, however MS demands thorough revision.
Comments:
Abstract: Abstract can be more concise. Add/change some keywords.
Introduction: There are two major concerns with this MS. First one is grammatical mistakes, language error, typographical mistakes. Second, the authors did not conceive the strong idea from review literature. Paragraphs and sentences did not have any link. Objectives of the study should be clearly defined.
Materials and methods: How many replications per treatment? How many plants per replication? Day/light hrs? Humidity? Temperature? Please elaborate detail method of “Soil organic matter content”.
Results and Discussion: In results, there is a striking lack of connectors between sentences and leading to confusing. I would suggest to present your results by increase/decrease %age. Percentage should be upto two digits e.g., 13% instead of 13.4%. One way of improving Discussion is to avoid repetition of results in this part. Discussion is very shallow and need in depth discussion with the recent literature published. In discussion, there is a lack of mechanistic approach. Spellings and English language needs to be checked thoroughly. Overall, drafting of many sentences need to be improved. Tidying up the text is also suggested.

---

## Round 0.2 · Minor Revisions

The paper is improved after revisions and may be accepted for publication. However, the language of the paper needs extensive revisions. Please improve the language of the paper and re-submit for further consideration.

**Language Note:** The Academic Editor has identified that the English language must be improved. PeerJ can provide language editing services - please contact us at copyediting@peerj.com for pricing (be sure to provide your manuscript number and title). Alternatively, you should make your own arrangements to improve the language quality and provide details in your response letter. – PeerJ Staff

Reviewer 1 ·

Basic reporting

no

Experimental design

no

Validity of the findings

no

Additional comments

no

·

Basic reporting

The paper on Assessing the effect of heavy metals on maize (Zea mays L.) growth and soil characteristics: plants implications for phytoremediation is written in good order now and the author incorporated all the comments carefully. It is according to the scientific standards and will attract researchers who are working on the influence of heavy metals on various field crops and soil properties. However, these heavy metals also interact with other pollutants, which negatively influence crop growth and physiology. But before going to publications, there are some modifications required to make it more accurate.

I suggest one paper that needs to be cited in this article that is most related and attracts the authors to explore the new insights further
Influence of soil microplastic contamination and cadmium toxicity on the growth, physiology, and root growth traits of Triticum aestivum L.


• Overall, I suggest accepting this manuscript

Experimental design

No comment

Validity of the findings

No comment

Additional comments

No comment

---

## Round 0.3 · accepted · Accept

The paper is improved after English editing and incorporating all comments of reviewers. Therefore, It is accepted for publication.